# Influence of different types of ionospheric disturbances on GPS signals at polar latitudes

Vladimir B. Belakhovsky[1], Yaqi Jin[2], Wojciech J. Miloch[2]

[1] Polar Geophysical Institute, Apatity, Russia
[2] Department of Physics, University of Oslo, Oslo, Norway

*Correspondence to*: Vladimir B. Belakhovsky (belakhov@mail.ru)

**Abstract.** The comparative research of the influence of different types of auroral particle precipitations and polar cap patches (PCPs) on the GPS signals disturbances in the polar ionosphere was done. For this purpose, we use the GPS scintillation receivers at Ny-Ålesund and Skibotn, operated by the University of Oslo. The presence of the auroral particle precipitation and polar cap patches was determined by using data from the EISCAT 42m radar on Svalbard. The optical aurora observations in 557.7 nm, 630.0 nm spectrum lines on Svalbard were used as well for the detection of ionospheric disturbances. The cusp identification was done with using SuperDARN (Hankasalmi) data.

We consider events when the simultaneous EISCAT 42m and GPS data were available for the 2010-2017 years, in paper we present in detail typical examples describing the overall picture and we present the statistics for 120 events. It was considered the dayside/cusp precipitation, substorm precipitations, daytime and nighttime PCPs, precipitation associated with the interplanetary shock wave arrival. We demonstrate that substorm-associated precipitations (even without PCPs) can lead to a strong GPS phase ($\sigma_\phi$) scintillations up to ~1.5-3 radians which is much stronger than those usually produced by other types of the considered ionosphere disturbances. The value of the substorm phase scintillations in general correlate with the value of the geomagnetic field disturbance. But sometimes even small geomagnetic substorm when it combined with the PCPs produces quite strong phase scintillations. Cusp phase scintillations are lower than dayside PCPs scintillations. PCPs can lead to stronger ROT (rate of total electron content) variations than other types of the ionosphere disturbances. So our observations suggest that the substorms and PCPs, being different types of the high-latitude disturbances, lead to the development of different types and scales of ionospheric irregularities.

## 1. Introduction

The Global Navigation Satellite Systems (GNSS) have a great influence on human society today. The ionosphere as a medium for the radio waves propagation can produce a negative impact on the quality of the received signal. Global

Positioning System (GPS) uses two frequencies: $f1 = 1575.42$ MHz and $f2 = 1227.60$ MHz. There are a lot of dual frequency GPS receivers all over the world which are used for the ionosphere studies. Irregularities in the plasma density can lead to rapid fluctuations of amplitude and phase of the signal which are known to as ionosphere scintillations [Basu et al., 2002]. Strong scintillations reduce the quality of the signal and even lead to the signal loss. Thus, the investigation of GPS scintillations is an important aspect of space weather. The level of scintillations is characterized by the phase ($\sigma\Phi$) and amplitude (S4) scintillation indices. Amplitude scintillations are caused by the plasma irregularities with scale sizes ranging from tens to hundreds of meters, while the phase scintillations are caused by the irregularities with the sizes from hundreds of meters to several kilometers. Ionospheric scintillations are most severe in the equatorial region and at high latitudes [Basu et al., 2002; Kintner et al., 2007].

The phase scintillations index is determined mainly by the diffractive (stochastic) GPS signal variations. In order to calculate scintillation indices a long-term trend caused by the satellite motion in relation to the receiver and ionosphere changes needs to be removed. A standard cutoff frequency (0.1 Hz) is commonly used for signal detrending since (Fremouw et al., 1978). This cutoff frequency is adequate for the equatorial and midlatitude ionosphere. But the high-latitude ionosphere is characterized by the high and variable ionospheric drift velocity (~100 m/s–1 500 m/s) which add refractive variations to the phase scintillation index. The value of the cutoff frequency affects on the phase scintillation index. So it often leads to the strong phase scintillations without amplitude scintillation. Some researchers to solve this problem introduce new scintillations indexes: Mushini et al. (2012) introduced improved phase-scintillation index ($\sigma_{chain}$), Forte (2005) introduced S$\varphi$ index. The fast iterative filtering signal decomposition technique was used to find optimal cutoff frequency (Ghobadi et al., 2020; Spogli et al., 2021). De Franceschi et al. (2009) have demonstrated the difficulty in finding the optimum cutoff frequency for statistical studies and they proposed to look for actual scintillations investigating the simultaneous occurrence of S4 and $\sigma_{\varphi}$ increase.

The high-latitude ionosphere is very dynamic and unpredictable structure forming plasma irregularities on a wide variety of scale sizes (from 1000 km to decameter scale). Case studies have indicated a relationship between auroral appearance and GPS scintillations (Smith, 2008; Prikryl et al., 2010, Kinrade et al., 2013). Mostly discrete aurora cause strong scintillations and cycle slips (i.e. jump in differential phase TEC). Phase scintillations are more prominent than amplitude scintillation in the polar ionosphere (Prikryl et al., 2010). Ionospheric phase disturbances over auroral region are frequent phenomena which often occur during night time (Prikryl et al., 2011).

One of the most dynamic ionospheric disturbances at high latitudes is the substorm. Magnetosphere substorm is a transient process originated on the nightside of the Earth, when a significant amount of energy derived from the solar wind-magnetosphere interaction is stored in the magnetosphere tail and then released into the auroral ionosphere (Rostoker et al., 1980). The substorms are characterized by the sporadic precipitations of energetic electrons from the magnetotail that lead to the appearance of the bright aurora. GPS signal phase scintillations correlate well with substorm-associated auroral disturbances.

It was found that polar moving auroral forms (PMAF) which are discrete auroral forms produced by the transient magnetopause reconnection have big impact on GNSS phase scintillations (Oksavik et al., 2015). Authors found that the scintillation impact is strongest when appearance of the PMAFs coincide with the appearance of the PCPs. Studies of (Van der Meeren et al., 2016) showed the sun-aligned arc in the polar cap does not cause significant phase scintillations.

At polar latitudes polar cap patches can produce severe ionosphere disturbances. Polar cap patches are 100–1000 km islands
of enhanced F-region plasma density. The patches are appeared near the cusp and then propagate along the ionospheric convection streams into the nightside auroral oval modulated by the nightside tail reconnection, after that patches returned to the dayside (Lockwood and Carlson, 1992; Zhang et al., 2013). During strong and stable polar cap convection, segmentation may not happen and a continuous tongue of ionization (TOI) may be formed across the polar cap [Foster et al., 2005]. Phase scintillations in TOI were studied in the paper (Van der Meeren, 2014), it was observed bursts of phase scintillations and no
amplitude scintillations in relation to the leading gradient of the TOI. Patches may develop smaller-scale irregularities down to decameter scale through the Kelvin-Helmholtz (KH) and gradient drift instabilities (Oksavik et al., 2012; Clausen et al., 2016). In the auroral oval, polar cap patches are termed auroral blobs (Lorentzen et al., 2010). Several types of auroral blobs are selected in previous investigations: boundary blobs, subauroral blobs, and auroral blobs (Crowley et al., 2000). These airglow emissions are detectable from ground optical instruments as structures propagating from high to low latitudes.

Statistical studies found one peak in the occurrence rate of GPS phase scintillation around magnetic noon and another peak around magnetic midnight (Jin er al., 2015). It is also found that the phase scintillation occurrence rate is higher in the dayside, while the strong phase scintillation occurs more frequently during night. Nonetheless, strong phase scintillation events can also be triggered at daytime in association with the PMAF.

Jin et al. (2014) found that polar cap patches have their biggest impact on GPS signals once they reach the nightside auroral
oval, in particular when combined with upward field-aligned currents (Clausen et al. 2016). Jin et al. [2014] focused on phase scintillation measurements inside polar cap patches identified in airglow imager data and found that patches have a moderate scintillation impact ($\sigma\phi \sim 0.2$). They also found, however, that the strongest impact on scintillations occurred when these patches cross from the polar cap into the auroral oval to become auroral blobs (Jin et al., 2016). It was shown in the papers (Jin et al., 2014; Jin et al., 2016) that PCPs can produce GPS scintillations quite comparable with scintillations during
the particle precipitations with appearance of strong green aurora. But it was a case study therefore this question needs further studies.

The high-latitude disturbances have a great negative influence of radio waves propagation. So it is important to know what type of the high-latitude ionosphere disturbances has strongest influence on GPS navigation systems. In the present work we address the following question: what disturbances in the polar ionosphere (particle precipitations or polar cap patches) have
stronger impact on the scintillations of GPS signals?

## 2. Instruments


The Ny-Ålesund (NYA) GPS scintillation receiver of the University of Oslo (UiO) was the main instrument used in our study. Upon availability of data, the Skibotn (Norway, mainland) GPS receiver was also used. The UiO GNSS scintillation receiver is the standard GNSS Ionospheric Scintillation/total electron content (TEC) Monitor (GISTM), model GSV4004B (Van Dierendonck et al., 1993). The carrier phase and power at the L1 frequency (1.57542 GHz) are tracked and recorded at

50 Hz rate. The phase ($\sigma\phi$) and amplitude scintillation indices (S4) are also calculated and recorded automatically. The phase scintillation index is defined as the standard deviation of the carrier phase that has been detrended by the high-pass sixth-order Butterworth filter with a cutoff frequency of 0.1 Hz. The amplitude scintillation index (S4) is defined as the standard deviation of the received signal power, based on the 50-Hz sampling rate, normalized to the average signal power over one minute periods.

In our paper we use the standard detrending (0.1 Hz) in order to be able to compare our results with previous results obtained with using this GPS receiver on Svalbard (Clausen et al., 2016; Jin et al., 2015, 2016, 2017, 2018). The choice of the fixed cutoff frequency can add refractive variations to the phase scintillation index. So term "phase scintillation index" used in our study means the phase fluctuations due to the presence of large-scale irregularities (above the Fresnel radius).

The GPS TEC data have been postprocessed by using WinTEC-P [Carrano et al., 2009]. The ROT (Rate of TEC) data over 1

min are also used to depict the TEC variations [see, e.g., Alfonsi et al., 2011], where ROT = $\Delta$TEC/$\Delta$t.

For the describing the ionospheric plasma parameters (density, ion and electron temperature, line of sight ion velocity as a function of range) we used the Svalbard EISCAT 42m radar and UHF radar in Tromsø. The beam of the EISCAT 42m radar is directed along the geomagnetic field (azimuth = 184°, elevation = 82°). The UHF radar beam was fixed at an azimuthal angle of 185° and elevation angle of 77° (i.e., in the field-aligned direction in the F region).

For some convenient cases the optical aurora observations performed by all-sky imager of University of Oslo, Polar Geophysical Institute on Svalbard was used. The ASI in Longyearbyen (named LYR5) is located at Kjell Henriksen Observatory (KHO; 78.2°N, 16.0°E; 75.4° MLAT). The LYR5 ASI uses an ICCD (intensified charge-coupled device). Imager record emission intensity across the sky with a 180° field of view fish eye lens at 630.0 nm every 30 s and 557.7 nm every 15 s, respectively. The intensity of ASI is calibrated into the standard kilo Rayleigh (kR) intensity scale.

The stations mentioned in this study have the following geographic and geomagnetic (CGM) coordinates:

Ny Ålesund (NYA) – [78.92°N, 11.95°E], [75.25°N, 112.08°E];

Longyearbyen (LYR) – [78.20°N, 15.82°E], [75.12°N, 113.00°E];

Hornsund (HOR) – [77.00°N, 15.60°E], [74.13°N, 109.59°E];

Tromsø (TRO) – [69.66°N, 18.94°E], [66.64°N, 102.90°E];

Skibotn (SKN) – [69.43°N, 20.38°E], [66.28°N, 103.41°E].

IMAGE magnetometers data were used for the geomagnetic field observations. OMNI database was used for the evaluating the solar wind and interplanetary magnetic field parameters. We use the 1 min resolution Sym-H index and AE indices to

show the geomagnetic storm and substorm activity. The Sym-H and AE indices were obtained from the World Data Center for Geomagnetism at Kyoto University (http://wdc.kugi.kyoto-u.ac.jp/).


### 3.  Analysis of ionosphere disturbances

In the present study we consider the influence of different geophysical phenomena on the GPS scintillations: dayside/cusp precipitations, nighttime substorm precipitation, daytime and nighttime polar cap patches, precipitations associated with the
interplanetary shock arrival. We focused mainly on the phase scintillation index because amplitude scintillation index (S4) practically has no significant variations at high latitudes. The presence of the particle precipitation into the ionosphere associated with the appearance of the "green line" aurora was determined as the density increase between 100-200 km altitudes according to the EISCAT radar data. The presence of the polar cap patches was determined as a strong density increase above 200 km altitude. We consider mainly the winter time because the EISCAT radar works most often in winter
season. We present 120 different cases for 2010-2017 years when the data from the EISCAT 42m radar was available and in this paper we analyze in detail typical examples. The presented conclusions are valid for the common picture. The preliminary analyze was done in the proceeding paper [Belakhovsky et al., 2020].

### 3.1. Dayside/cusp precipitations

Example of the dayside precipitations on 9 January 2016 is shown on Figure 1. It was geomagnetically quite period (SYM-H
= -10 nT), solar wind speed had moderate values (V = 460 km/s). At time intervals 04-08 UT and 12-14 UT it is seen the ionosphere density increase at altitudes 100-200 km associated with the charge particle precipitation according to the EISCAT 42m data (MLT = UT + 3). The increase of the aurora intensity in 557.7 nm emission line measured by the NYA all-sky imager testifies about presence of the particle precipitation as well (Fig. 2.).

During this geomagnetic conditions NYA station often was located near the cusp region. Cusp is a small area near noon, it
reaches about 3 MLT in longitude and some degrees in latitudes. The cusp can be identified by the spectral width of reflected ionosphere signal. In the region of the cusp, an increase in the spectral width of the reflected signal is observed due to an increase of the turbulence. Cusp is observed with an increase in the spectral width of the reflected signal up to 200 m/s at near noon hours. The methodic of the cusp identification with using SuperDARN data was done for example in (André et al., 2000l Pilipenko et al., 2017). According to the SuperDARN radar (beam 9) observations in Hankasalmi (Figure 2) the cusp
near the Svalbard was registered approximately from 08 to 12 UT. The equatorial boundary of the cusp was located at 78 degrees of geographic latitudes and higher. The growth of the aurora intensity in 630.0 nm emission lines at 07-09 UT during low intensity in 557.7 nm may also testifies about the optical identification of the cusp.

The different colors on panels 2-5 (Figure 1) are data from the different GPS satellites. The GPS phase scintillation index reaches the value about 0.4-0.5 radian. Possibly the jumps in phase index variations in time interval 08-11 UT may testify
about the passing of the station near the cusp. There is no strong Ne increase below 200 km, testifying about the particle

precipitations, according to the EISCAT 42m data at 08-11.30 UT. At the moment of the passing under the cusp region the VTEC (up to 12 TECU), ROT (up to 5 TECU/min) increase was also observed (Figure 1).

The growth of the phase index was seen not during all period of Ne enhancement (100-200 km) measured by the EISCAT radar. Possibly this is due to the field of view of EISCAT radar not coincides with the field of view of GPS receivers. So for small-scale and medium-scale Ne increase is not always correlate with phase index growth.

The ROT variations have small disturbances (2-4 TECU/min) during the presence of the particle precipitation. Amplitude scintillation index have no distinct response to the morning and daytime precipitations.

It was not observed substorm disturbances during this day. It is seen small amplitude Pc5 pulsations (10-20 nT) in X-component of the geomagnetic field at NYA station during morning precipitations. These Pc5 pulsations can have contribution to the charge particle precipitation into the ionosphere (Belakhovsky et al., 2016).

## 3.2. Substorm precipitations

The example of the substorm precipitations and the GPS scintillations response to it is shown in Figure 3 (11 December 2015). It was observed two substorms during this day. The first one was at 15.30-17.00 UT, the second was at 20.00-22.00 UT. It was polar substorms because it mainly observed at latitude higher than 70º. It can be noticed that the amplitude of the first substorm reaches the value about 1400 nT at Hornsund (HOR) station, at NYA station the amplitude of the substorm was 600 nT. The second substorm has lower amplitude than first substorm (600 nT at HOR station). These substorms was observed without geomagnetic storm (SYM-H $\approx$ - 10 nT), however the solar wind speed was quite high (V = 640-680 km/s) according to the OMNI database (data not shown).

The ionosphere plasma density $\Delta$Ne increases during substorm on two orders from $1 \cdot 10^4$ to $1 \cdot 10^6$ cm$^{-3}$ at altitude about 104 km, at altitude 125 km $\Delta$Ne increases from $1 \cdot 10^4$ to $5 \cdot 10^5$ cm$^{-3}$. The substorms were accompanied by the strong increase of aurora intensity in different spectrum lines. The growth of the phase scintillation index was accompanied by the appearance of the bright aurora arc according to LYR all-sky camera observations (Figure 4) oriented approximately in east-west direction. We have plotted ionosphere pierce points (IPP) to show how it overlaps with the EISCAT 42m and all-sky camera field of view (Figure 4). For large-scale disturbances (hundreds of km) which were considered in this paper this overlap does not matter. Always some GPS satellites can be used to register the ionosphere disturbance.

The phase index reaches the value about 2 radians during the first substorm. The growth of the phase scintillation index was seen mainly during the substorm expansion phase (30-40 minutes). During the second substorm the phase scintillation index has the lower value (0.5-1.5 radians). The growth of the phase scintillation index was seen as sharp increases during 5-10 minutes time intervals.

Substorms do not lead to significant TEC increase. It is seen the absence of the TEC data during the substorm. It testifies about the cycle slips of GPS signal at this moment.

The ULF waves in Pi3 frequency range embedded into the substorm structure have contribution into the particle acceleration into the ionosphere, growth of the field-aligned currents (FAC) which leads to such strong values of the phase scintillation index.

### 3.3. Precipitations associated with the interplanetary shock arrival

It is considered the example of the interplanetary shock influence on scintillations of GPS signals. Here the example for the event of 22 January 2012 is presented. It is well known that the interplanetary shock interaction with the Earth's magnetosphere leads to strong particle precipitation into the ionosphere and leads to the appearance of the so named shock-aurora (Zhou et al., 2003). The interplanetary shock interaction with the Earth magnetosphere which leads to the development of the geomagnetic storm is known as SSC (storm sudden commencement) otherwise when there is no geomagnetic storm it is named as SI (sudden impulse). In the paper (Belakhovsky et al., 2017) it is shown that shock wave arrival produces to the global TEC increase at high latitudes up to 40%.

The considered interplanetary shock is accompanied by an abrupt increase of the solar wind velocity (from 320 to 400 km/s), density (from 20 to 40 cm$^{-3}$), temperature, magnitude of the interplanetary magnetic field (from 12 to 25 nT) according to the OMNI database, abrupt growth of the SYM-H index at 06.10 UT (Figure 5). The NYA station at the moment of interplanetary shock arrival was located on the morning side (09 MLT). For the SSC event the phase index reaches the value near about 0.3-0.4 radians (Figure 6). The ROT reaches the value about 4 TECU/min. So SSC event does not lead to the strong GPS scintillations.

For the considered interplanetary shock cases the phase index reaches similar values (less than 1 radian).

### 3.4. Polar cap patches

***Nighttime polar cap patches***. The example of the evening-nighttime polar cap patches (PCPs) is shown on Figure 7 (left panel) for the 10 February 2015. The PCPs were observed at 19.00-23.30 UT as a density increase above 200 km according to the EISCAT 42m radar data. The ionosphere plasma density ΔNe increases during PCPs from $8 \cdot 10^4$ to $7 \cdot 10^5$ cm$^{-3}$ at altitude about 321 km. At NYA GPS receiver the phase scintillation index reaches the medium value (0.4 radians). However the ROT variations for the PCPs reach the high values (10-15 TECU/min).

During the PCPs appearance the Bz-component of the IMF has negative values (-6 nT) during 3 hours (data not shown). It leads to the development of the small substorm. The amplitude of the substorm is 120-140 nT in X-component of the geomagnetic field at NYA station. The PCPs is also identified in the aurora intensity variations as forms propagating from the polar to low latitudes in 630.0 nm (red line) emission (Figure 8) at 19.00-23.00 UT according to LYR all-sky camera observations.

It was done the comparison of the PCPs development on EISCAT 42m radar (Svalbard) and EISCAT UHF radar located at lower latitudes, in Tromsø (Figure 7, right panel ). The sharp increase of the plasma density (Figure 8) from 12 to 17 UT

above 200 km is caused by the sunlight. During appearance of PCPs near the Svalbard at latitudes of the SKN (TRO) stations a long lasting substorm (more than 4 hours duration) with the amplitude 200-250 nT was observed. This substorm produces strong $\Delta$Ne increase below 200 km and at altitudes 200-550 km (Figure 7, right panel). At SKN station the phase scintillation index has approximately the same values (0.4-0.5 radians) as in polar latitudes (NYA station) but for one moment $\sigma_\Phi$ reaches the value about 0.7 radians. ROT variations at SKN station have lower values (6 TECU/min) than on Svalbard.

***Daytime polar cap patches***. It was also analyzed the influence of polar cap patches which was observed on the dayside on GPS signals scintillations. It is considered the geomagnetic storm on 22 January 2012 (Figure 5, Figure 6). The SSC event for this storm was considered in section 3.3 of this paper. It was CME (coronal mass ejection) geomagnetic storm conditions, the SYM-H index reached the value about -80 nT, the solar wind speed has maximum about 480 km/s according to the OMNI database.

According to the ionosphere plasma density measurements on EISCAT 42m radar the PCPs were registered after the development of the geomagnetic storm, in time interval 08.00-12.00 UT (11-15 MLT) – Figure 6. The ionosphere plasma density $\Delta$Ne increases during PCPs from $5 \cdot 10^4$ to $7 \cdot 10^5$ cm$^{-3}$ at altitude about 321 km. The negative bay in geomagnetic field variations with amplitude about 300 nT is seen on NYA station.

During this event the strong GPS phase scintillations (0.5-0.8 radians) were registered in whole time interval of the PCPs appearance. Amplitude S4 index has no any clear response to the PCPs. It was observed VTEC increase (from 5 to 18 TECU) caused by the PCPs contribution. ROT variations have high values (7 TECU/min).

For the all of the considered PCPs cases phase index has the value less than 1.

## 3.5. Statistical analysis

To confirm statistically results of our case studies we present 4 tables which include values of phase scintillation index, ROT values for the considered types of the ionosphere disturbances. Table 2 and Table 3 also include magnitude of the geomagnetic field disturbance (X-component) for the substorms, SSC/SI events. Often during one day some types of the ionosphere disturbances was observed.

Table 1 presents 33 dayside/cusp precipitations events: phase scintillation index is less than 1 radian with the medium values about 0.5-0.6 radian. ROT values are no more than 6 TECU/min.

Table 2 presents 35 substorm events. For many substorm events phase scintillation index is more than 1, ROT values are no more than 6 TECU/min. The maximum value of the phase scintillations are 2.9 radian fixed for the 20 January 2010 event when the geomagnetic field disturbance $\Delta$X at NYA stations was about 750 nT. We also select separately substorms which were observed during the PCPs appearance. It is seen from the substorm statistics that in general phase index increase correlate with the level of the geomagnetic disturbance. But sometimes even small amplitude substorms lead to the

significant values of the phase index when it combined with the PCPs. Possibly presence of the PCPs give growth to the level of scintillations (Jin et al., 2014; Clausen et al. 2016).

Table 3 presents 14 SSC/SI events. Shock induced precipitations lead to medium values of the phase scintillation index (0.4-0.5 radians), and medium values of the ROT (4-6 TECU/min). SSC events when it was observed on nighttime often trigger the development of the geomagnetic substorm. So high values of the phase scintillations (near 1 radian) can be caused by the triggering of the substorm immediately after SSC. Often there is no significant increase of the GPS phase scintillations was observed after the SSC.

Table 4 shows 38 daytime and nighttime PCPs events. It is seen from the Table 3 that PCPs are less than 1. Analysis of the tables 3 shows that PCPs produce more strong ROT variations (even >10 TECU/min) than other types of the polar latitude ionosphere disturbances.

Jin et al. (2017) investigated the GPS scintillations around cusp region and found that cusp precipitations have stronger influence on GPS phase scintillation when it combined with the PCPs. Our research shows that daytime PCPs can produce

stronger GPS phase scintillations than dayside/cusp precipitations. See Table 1 and Table 4.

It is also seen from the Tables that often one type of the ionosphere disturbances was registered during several consecutive days. For example for the PCPs it is 28-30 November 2011, 4-6 November 2013, 20-26 January 2014, for the substorms it is 10-13 December 2015, for the cusp/daytime precipitation it is 17-19 December 2014. Of course there are more events with significant increase of the phase scintillations during 2010-2017 years. But we pay attention mainly to the events when the

GPS and EISCAT data were available simultaneously to precisely define the physical nature of the ionosphere disturbance. Only for the SSC/SI events we not always used the EISCAT radar data. So analysis of the group of the events confirms in general the detail analysis of the individual events.

## 4.    Discussion

The high-latitude ionosphere is a very dynamic structure due to the charge particles (mainly electrons and protons) penetrating from the outer space which can lead to the appearance of the different times-scale ionosphere irregularities. Variations of the ionospheric electron density cause variable group delay and phase advance of the radio wave, resulting in rapid phase fluctuations or phase scintillations. The scintillations of the GPS radio signal are caused by refraction and

diffraction of radio waves passing through ionospheric irregularities on scales from tens of meters to a few kilometers (Basu et al., 1998; Kintner et al., 2007). There is a problem of the finding of optimal cutoff frequency during phase detrending. The choice of the standard cutoff frequency (0.1 Hz) can lead to the presence of the refractive variations in phase scintillation index (Forte, 2005; Mushini et al., 2012; Ghobadi et al., 2020; Spogli et al., 2021).

There are a lot of disturbances at polar ionosphere. The main phenomena are substorm and polar cap patches. In this work it

is considered the influence of different types of the high latitude ionosphere disturbances (such as dayside/cusp precipitation, substorms, nighttime and daytime polar cap patches, interplanetary shock wave) on perturbations of GPS signals with using

GPS scintillation receiver on Svalbard. For some events also SKN GPS scintillation receiver on mainland (Norway) was used. In this paper we present case studies and statistical analysis.

Concerning amplitude GPS scintillations at high latitudes we confirm the previous findings and did not found any certain reaction of the S4 index to the considered disturbances. However, Jin et al. (2018) have found S4 scintillations for the very strong geomagnetic storm on 17 March 2017. But it was extreme event. For the ordinary events S4 index at high latitudes does not experience great increases. Possibly low values of amplitude scintillations at high latitudes are caused by the low elevation angles of GPS satellites at these regions. Since irregularities producing amplitude scintillations can be formed in the field-aligned direction. But this hypothesis needs to be tested.

The analysis shows that the polar substorms even if it observed without PCPs lead to the maximum values of the phase scintillation index (1.5-3 radians). From one side it is obvious because the substorm is a most powerful disturbance in the magnetosphere-ionosphere system, it leads to the growth of the ionosphere plasma density on two orders. The ULF waves (Pi3 pulsations) embedded into substorm can accelerate electron along the geomagnetic field lines and produce auroral arcs. The duration of the substorm is about 1.5-3 hours while the growth of the phase scintillation index was observed during 30-40 minute (expansion phase). So the substorm leads to the high but short-time growth of the phase index. In paper (Kim et al., 2014) it is shown that ULF waves in Pi2 frequency range have a dominant role in producing accelerated auroral electrons. During the substorms bright and discrete auroral forms are appeared on the sky. Such inhomogeneous ionization structures produce significant changes in the refractive index and enhance the phase scintillation index (Hosokawa, 2014). In addition, field-aligned currents, produced field-aligned irregularities, can be the major driver of high-latitude ionospheric irregularities (Prikryl et al., 2011).

The phase scintillations are typically produced by variations in the refractive index due to ionospheric irregularities of the scale from a few kilometers to a few tens of kilometers (Kintner et al. 2007). The phase scintillations during the substorm interval were produced through the refractive process caused by large-scale density irregularities associated with rapidly moving auroral arc. Hosokawa et al. (2014) found that phase scintillation was enhanced in relation to substorm onset and decreased as the aurora became more diffuse. They suggested that discrete aurora in the GPS signal path is necessary for the occurrence of phase scintillations during substorm intervals. In addition to scintillation, other effects such as loss of lock (Smith et al., 2008) and cycle slips (Prikryl et al., 2010) have been directly observed in relation to auroral emissions. The long-term statistical study (De Franceschi et al., 2019) over Svalbard for the 2013-2016 years shows that magnetic midnight region is the most exposed to scintillations.

Polar cap patches are the source of decameter to kilometer-scale irregularities causing scintillations. In some of the considered cases the polar cap patches were observed during 6 hours on EISCAT 42m radar. Analysis shows that PCPs are accompanied by the lower values of phase index than during a substorm but this growth is registered during longer time intervals. At the same time the PCPs lead to the substantially higher values of the ROT variations. ROT is the time rate of change of the differential carrier phases, it providing information about scale size of the electron density irregularities scale which produces GPS signal scintillations. We used in our study 1-minute ROT time resolution. The typical velocity of the

plasma convection at high latitudes is between 100 m/s and 1 km/s. So 1-minute ROT variations are caused by the irregularities with the scale about 6-60 km. Our observations suggest that the substorms and PCPs, being different types of the high-latitude disturbances, lead to the development of different types and scales of ionospheric irregularities.

Comparison of the EISCAT observations on Svalbard and in Tromso shows that during PCPs appearance on Svalbard a typical substorm at lower latitudes (Tromso) was observed. The level of the phase scintillations are quite comparable at high (Tromso) and polar (Svalbard) latitudes but the level of ROT is higher at polar latitudes.

There are two plasma instabilities which can explain the ionospheric irregularities at high latitudes: the gradient drift instability (GDI) and the Kelvin–Helmholtz instability (KHI). The GDI requires a density gradient and it can produce irregularities at the trailing edge of a plasma patch, while the KHI requires a velocity shear and the irregularities can be created around boundaries of velocity shears. The GDI can work on these sharp density gradients very efficiently (Moen et al., 2013) to produce small-scale irregularities which cause scintillations. The polar cap patches represent the largest scale structure in the high latitude ionosphere. During their convection in the polar cap, the Gradient Drift Instability (GDI) acts on them and develops small scale irregularities on the trailing edge.

The identifications of the cusp region with using the SuperDARN and all-sky cameras were done in our study. Cusp region is a source of the plasma turbulence of the different scales. It is found the medium growth of the TEC and ROT near the cusp region. Cusp phase scintillations are lower than dayside PCPs scintillations. On the dayside, the cusp is an active region of the GPS scintillation (Moen et al., 2013), where loss of signal locks occurs (Oksavik et al., 2015). By analyzing GPS phase scintillations around magnetic noon, it is suggested that GPS phase scintillations are sensitive to a combination of the cusp aurora and the intake of solar EUV-ionized plasma (Jin et al., 2015).

The common analysis shows that for all of the considered events the significant phase scintillations were observed. PCPs lead to the prolonged variations of phase index but with smaller values (less than 1). Shock induced precipitations, daytime and cusp precipitations lead to the medium values of phase index (0.4-0.5 radians) and medium values of the ROT. But among the all types of the disturbances the substorms leads to the greatest values of the phase scintillations index. Thus, the substorm precipitations have the strongest impact on the scintillations of GPS radio signals in the polar ionosphere even without PCPs. Substorm leads to short-time (10-20 minute) and strongest values of GPS phase scintillations while PCPs to long-time (some hours) medium values of GPS phase scintillations. So both types of these ionosphere disturbances are dangerous for the quality of communications, navigations, and locations at polar latitudes.

## 5. Conclusions

It is considered the influence of different types of the high latitude ionosphere disturbances (such as daytime/cusp precipitations, substorms, nighttime and daytime polar cap patches, interplanetary shock wave) on perturbations of GPS signals with using receivers on Svalbard and in Skibotn. All of the considering types of the ionospheric disturbances lead to the growth of phase scintillations index and ROT variations.

Substorms (even without PCPs) lead to the maximum values of the phase scintillation index (1.5-3 radians). The growth of the phase index observed mainly during the substorm expansion phase. ULF waves in Pi3 frequency range during a substorm producing auroral arcs can lead to such high values of the phase scintillation index.

The value of the substorm phase scintillations in general correlate with the value of the geomagnetic field disturbance. But sometimes even small geomagnetic substorms when it combined with the PCPs produce quite strong phase scintillations.

Polar cap patches lead to the prolonged variations of phase index with smaller values (less than 1). At the same time polar cap patches can lead to strong ROT variations (10-15 TECU/min) in comparison with the substorms disturbances. So our observations suggest that the substorms and PCPs, being different types of the high-latitude disturbances, lead to the development of different types and scales of ionospheric irregularities.

Cusp region, indentified with using SuperDARN radar, leads to the moderate growth of phase scintillation index and ROT variations. Shock induced precipitations lead to medium values of the phase scintillation index (0.4-0.5 radians) and medium values of the ROT (4-6 TECU/min). Our analyses shows that there is no clear response of the amplitude scintillation index to the different types of the ionosphere disturbances.

**Code and data availability**. The authors thank the Norwegian Polar Research Institute at Ny-Ålesund for assisting us with the GPS receiver in Ny-Ålesund, Bjørn Lybekk and Espen Trondsen for the instrument operations. The IMF data are provided by the NASA OMNI Web service (http://omniwegsfc.nasa.gov).

The authors wish to thank IMAGE (http://www.ava.fmi.fi/image/), EISCAT groups for the available data. EISCAT is an international association supported by research organizations in China (CRIRP), Finland (SA), Japan (NIPR and STEL), Norway (NFR), Sweden (VR), and the United Kingdom (NERC). Data from EISCAT can be obtained from the Madrigal database http://www.eiscat.se/madrigal. The University of Oslo ASI data are available at http://tid.uio.no/plasma/aurora.

The authors acknowledge the use of SuperDARN data. SuperDARN is a collection of radars funded by national scientific funding agencies of Australia, Canada, China, France, Italy, Japan, Norway, South Africa, United Kingdom and the United States of America.

**Author contributions**. VB take a part in formulation of the problem, analyze events, do conclusions, and write text with contributions from all the co-authors. YJ take a part in formulation of the problem, process GPS, optical data, correct text. WM take a part in formulation of the problem, organize this research team.

**Competing interests**. The authors declare that they have no conflict of interest.

**Acknowledgements**. This work is supported by the Russian Science Foundation (grant # 18-77-10018).

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

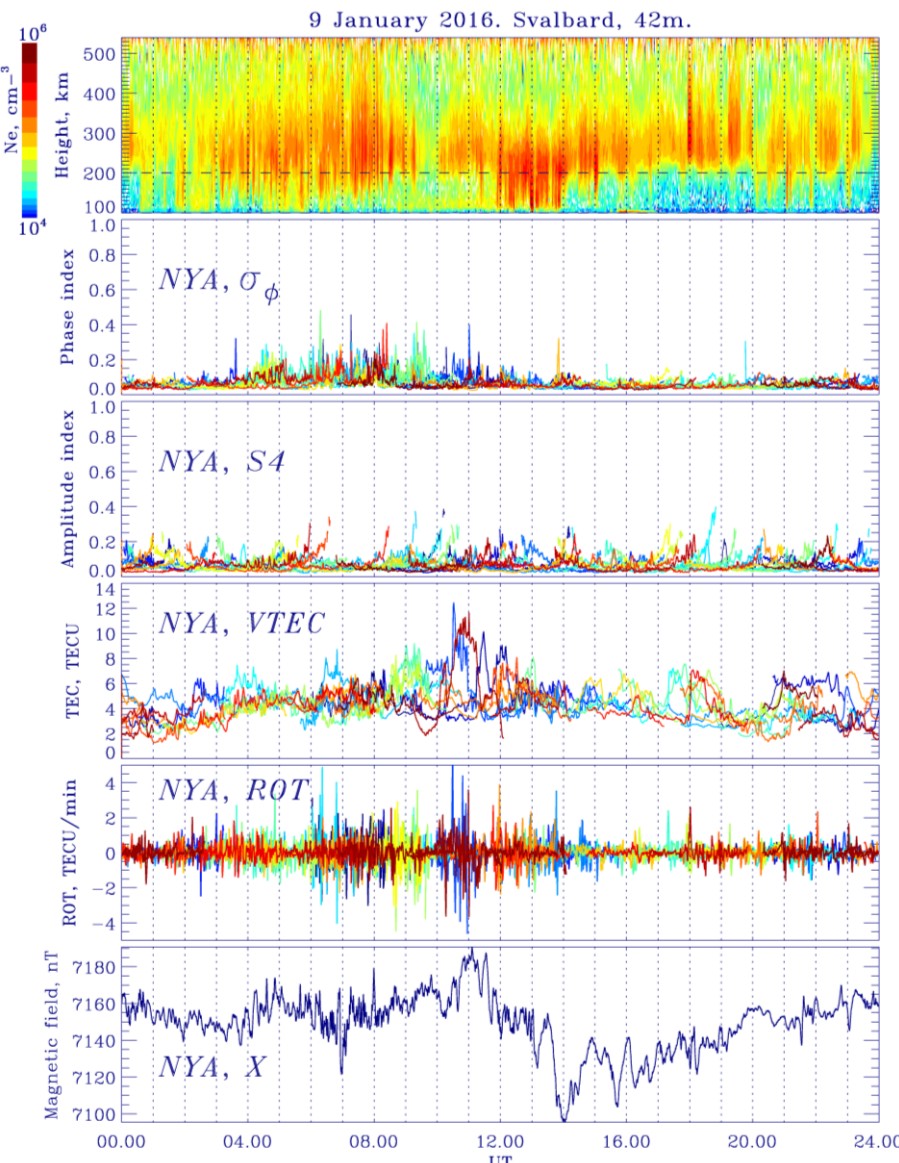

**Figure 1: Ionosphere plasma density according to the EISCAT 42m radar data; phase scintillation index and amplitude scintillation index according to the GPS receiver at NYA station; TEC and ROT variations according to the GPS receiver at NYA station; geomagnetic field variations (X-component) at NYA station for the 9 January 2016.**




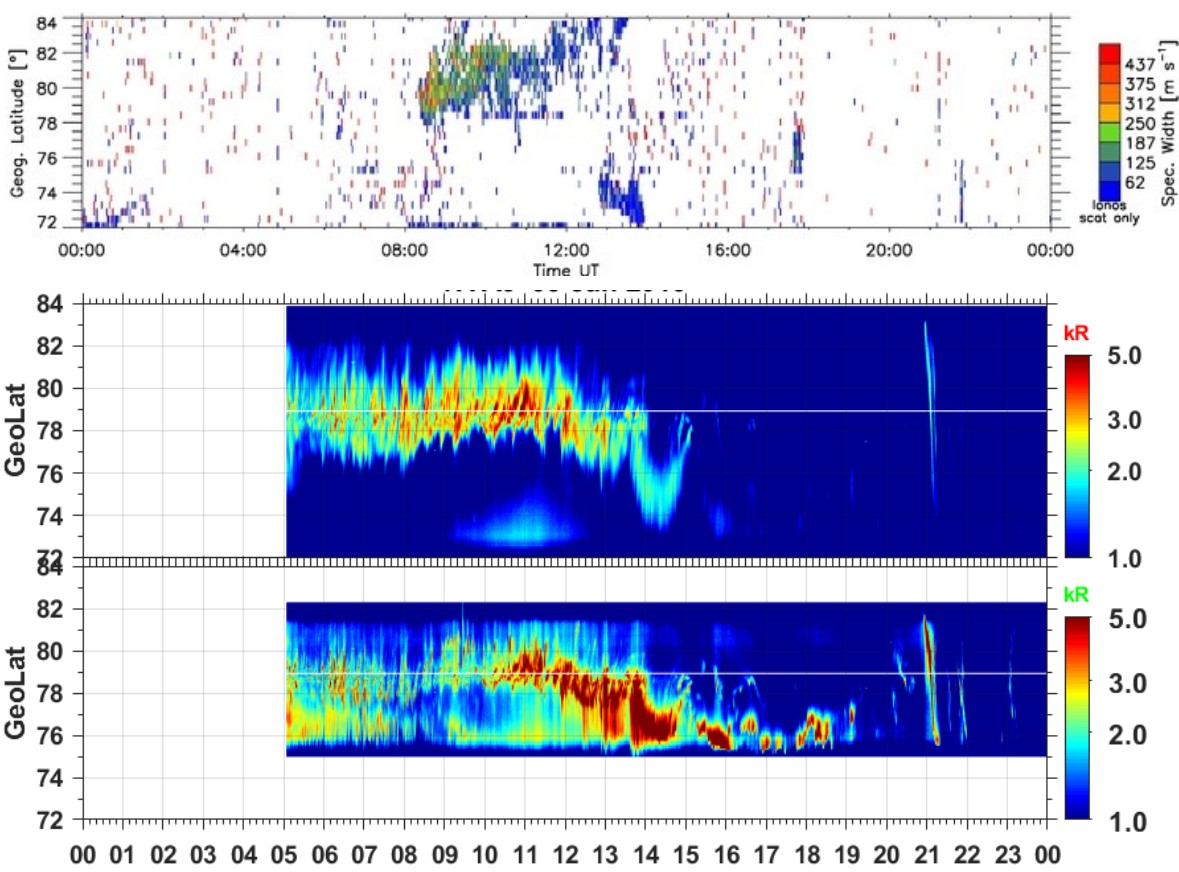

**Figure 2: Latitude-time plot of the spectral width according SuperDARN Hankasalmi radar, keograms (557.7 nm and 630.0 nm**
**emission line) from the all-sky imager at NYA station for the 9 January 2016.**


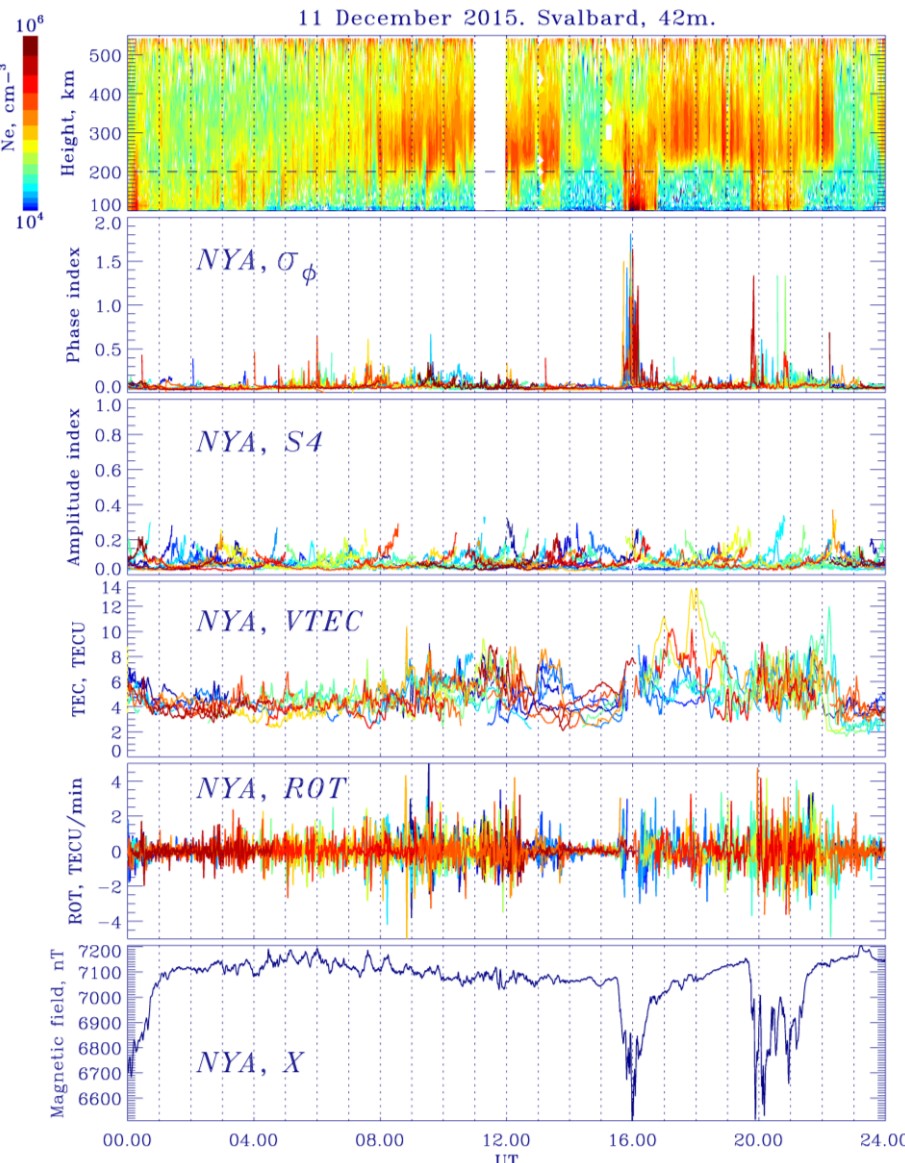

Figure 3: Ionosphere plasma density according to the EISCAT 42m radar data; phase scintillation index and amplitude scintillation index according to the GPS receiver at NYA station; TEC and ROT variations according to the GPS receiver at NYA station; geomagnetic field variations (X-component) at NYA station for the 11 December 2015.



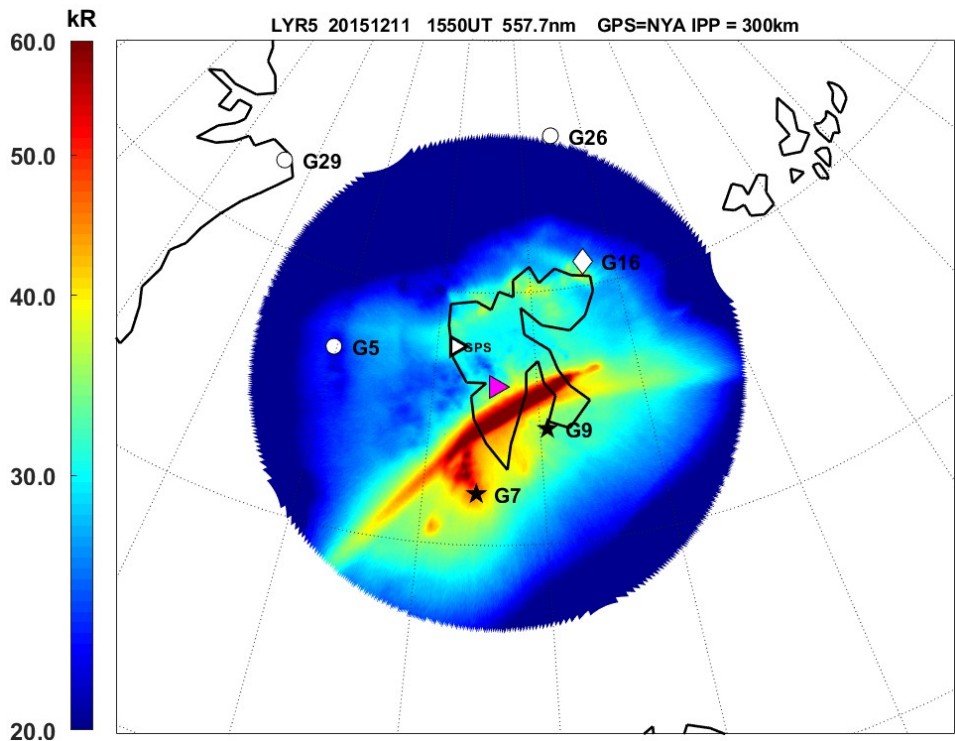

**Figure 4: The all-sky camera image (557.7 nm emission) at LYR station projected into the map for the 15.50 UT 11 December**
**2015. The rose triangle is a EISCAT 42m radar, the white triangle is a GPS receiver on NYA station, the G7, G9, G5, G16, G29,**
**G26 is a ionosphere pierce points of the corresponding GPS sattelites.**


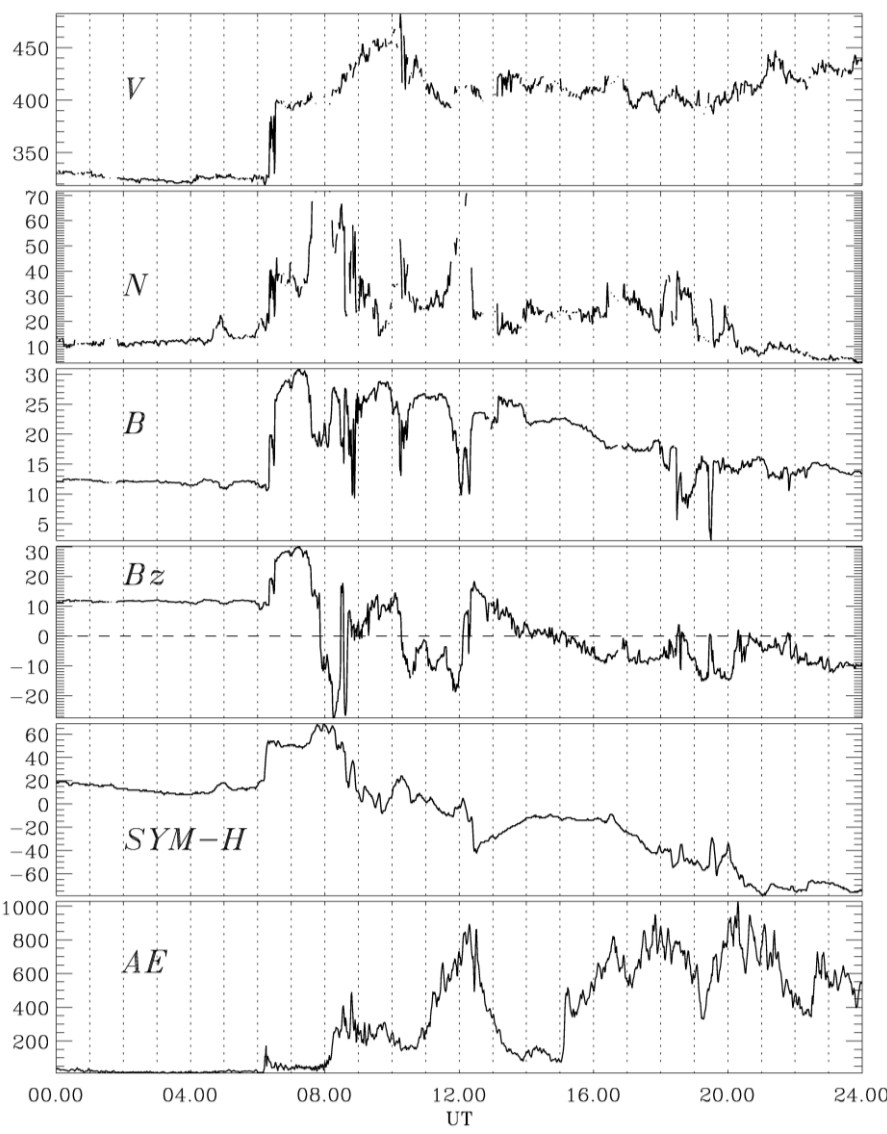

**Figure 5: Solar wind speed V [km/s], solar wind density N [cm-3], magnitude of the IMF B [nT], Bz-component [nT] of IMF according to the OMNI database, SYM-H index [nT], AE index [nT] for the 22 January 2012.**

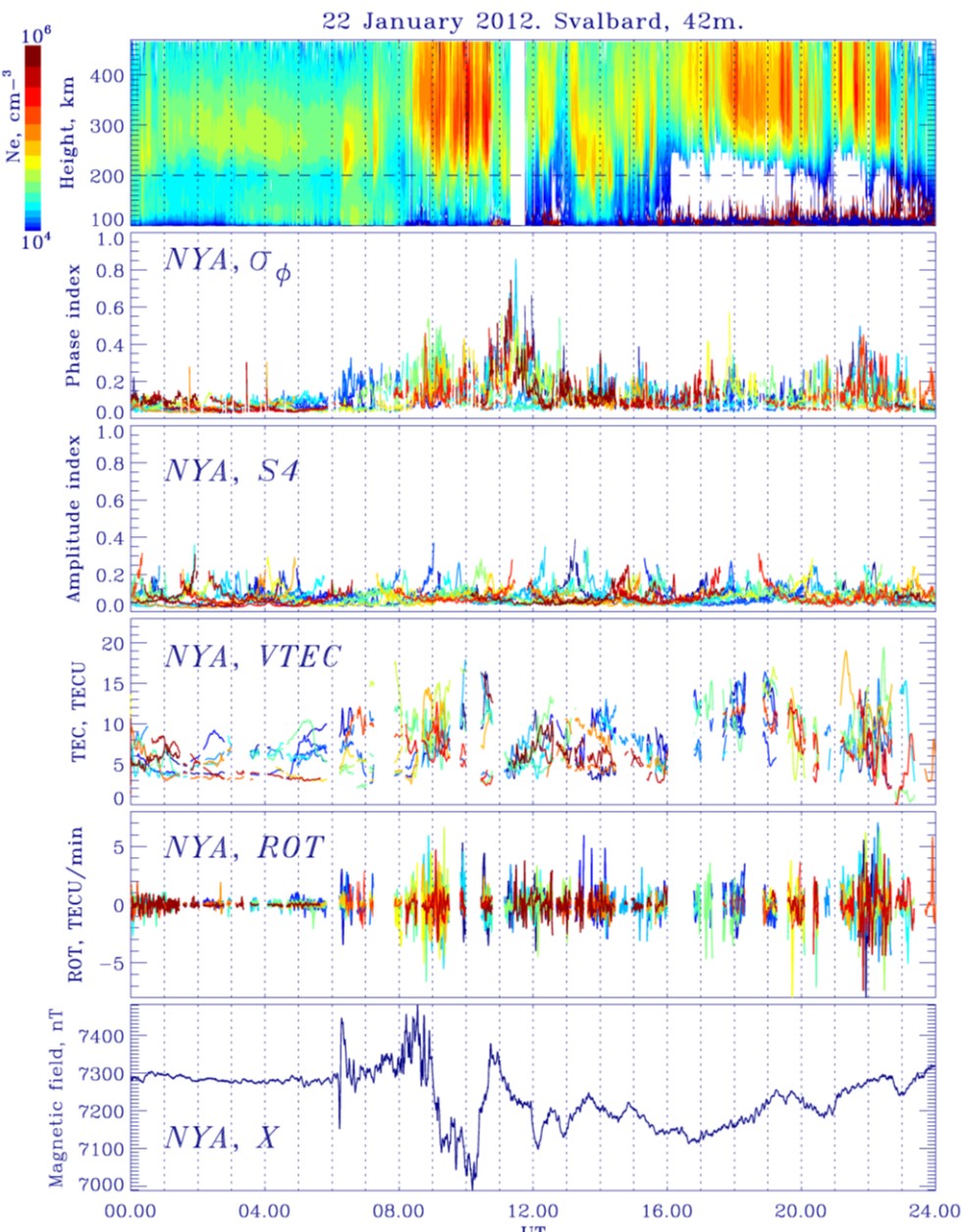

**Figure 6: Ionosphere plasma density according to the EISCAT 42m radar data; phase scintillation index and amplitude scintillation index according to the GPS receiver at NYA station; TEC and ROT variations according to the GPS receiver at NYA station; geomagnetic field variations (X-component) at NYA station for the 22 January 2012.**

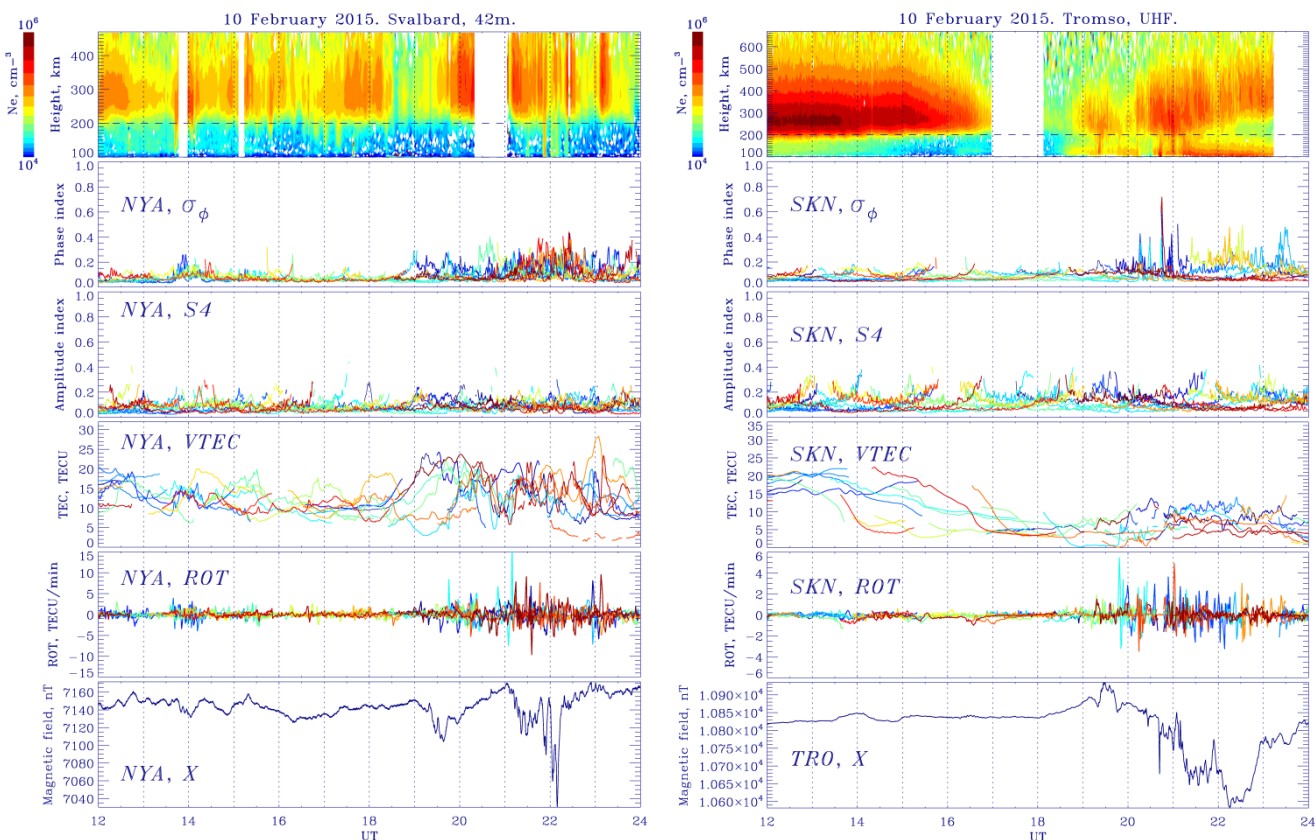

**Figure 7: Left panel: ionosphere plasma density according to the EISCAT 42m data; phase, amplitude scintillation indexes, TEC and ROT variations according to the GPS receiver at NYA station; X-component of the geomagnetic field at NYA station. Right panel: ionosphere plasma density according to the UHF EISCAT data; phase and amplitude scintillation indexes, TEC and ROT variations according to the GPS receiver at SKN station; X-component of the geomagnetic field at TRO station for the 10 February 2015.**


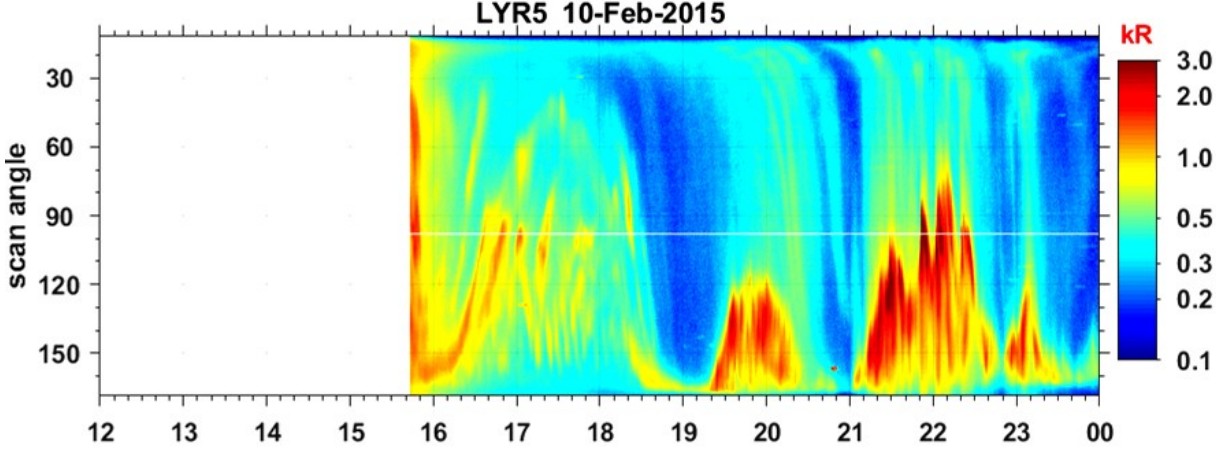

**Figure 8: Keogram (630.0 nm emission line) from the all-sky imager at LYR station for the 10 February 2015.**



**Table 1. The values of the phase scintillation index, ROT, determined form GPS receiver at NYA station during dayside/cusp precipitation events.**

| Date (day/month/ year) | Time, UT | Phase index, deg | ROT, TECU/ min |
|---|---|---|---|
| **Dayside/cusp precipitation** | | | |
| 16.01.2012 | 09.00-14.00 | 0.7 | 4 |
| 09.01.2014 | 05.30-11.00 | 0.5 | 6 |
| 20.01.2014 | 08.00-10.00 | 0.9 | 4 |
| 21.01.2014 | 05.00-09.00 | 0.6 | 4 |
| 23.01.2014 | 04.00-07.00 | 0.85 | 4 |
| 25.01.2014 | 04.00-08.00 | 0.35 | 4 |
| 26.01.2014 | 06.00-09.00 | 0.35 | 3 |
| 17.12.2014 | 06.00-12.00 | 0.5 | 4 |
| 18.12.2014 | 05.00-10.00 | 0.55 | 3 |
| 19.12.2014 | 06.00-11.00 | 0.55 | 5 |
| 10.12.2015 | 05.00-11.30 | 0.6 | 5 |
| 11.12.2015 | 05.00-12.00 | 0.7 | 4 |
| 13.12.2015 | 05.00-12.00 | 0.55 | 5 |
| 06.01.2016 | 05.00-12.00 | 0.55 | 3.5 |
| 07.01.2016 | 05.00-12.00 | 0.6 | 4 |
| 09.01.2016 | 04.00-12.00 | 0.5 | 6 |
| 10.01.2016 | 05.00-12.00 | 0.7 | 4 |
| 12.01.2016 | 06.00-12.00 | 0.8 | 5 |
| 12.01.2016 | 14.00-16.00 | 0.5 | 6 |
| 06.02.2016 | 08.00-13.00 | 0.3 | 6 |
| 07.02.2016 | 09.00-14.00 | 0.4 | 6 |
| 08.02.2016 | 07.00-15.00 | 0.4 | 4 |
| 10.02.2016 | 06.00-12.00 | 0.35 | 3 |
| 05.10.2016 | 04.00-08.00 | 0.8 | 4 |
| 27.10.2016 | 07.00-11.30 | 0.65 | 4 |
| 01.12.2016 | 04.00-10.00 | 0.35 | 4 |
| 20.01.2017 | 12.00-14.00 | 0.5 | 4 |
| 25.01.2017 | 06.00-12.00 | 0.8 | 5 |
| 01.02.2017 | 08.00-12.00 | 0.6 | 4 |
| 02.02.2017 | 08.00-12.00 | 0.6 | 4 |
| 04.02.2017 | 11.00-14.00 | 0.75 | 6 |
| 17.12.2017 | 07.00-13.00 | 0.9 | 4 |
| 18.12.2017 | 05.00-12.00 | 0.7 | 4 |

**Table 2. The values of the phase scintillation index, ROT, determined form GPS receiver at NYA station, and corresponding levels of the geomagnetic field disturbance (ΔX) during different substorm and substorm during PCPs events.**

| Date (day/month/year) | Time, UT | Phase index, deg | ROT, TECU/min | ΔX, nT |
|---|---|---|---|---|
| **Substorms** | | | | |
| 20.01.2010 | 17.00 | 2.9 | 4 | 750 |
| 01.12. 2011 | 02.30 | 1 | 3 | 300 |
| 01.12.2011 | 20.00 | 0.55 | 4 | 200 |
| 16.01.2012 | 21.00 | 1.15 | 4 | 300 |
| 09.01.2014 | 20.30 | 0.7 | 6 | 450 |
| 10.12.2015 | 17.00 | 1.8 | 6 | 550 |
| 10.12.2015 | 20.30 | 1 | 4 | 600 |
| 11.12.2015 | 16.00 | 1.8 | – | 600 |
| 11.12.2015 | 20.00 | 1.3 | 6 | 650 |
| 12.12.2015 | 21.30 | 1.1 | 2 | 180 |
| 12.12.2015 | 22.30 | 0.7 | 2 | 70 |
| 13.12.2015 | 21.00 | 1.3 | 2 | 120 |
| 06.01.2016 | 01.20 | 1.3 | 4 | 700 |
| 07.01.2016 | 20.00 | 0.75 | 4 | 250 |
| 10.01.2016 | 21.00 | 0.75 | 2 | 180 |
| 05.02.2016 | 19.30 | 1.2 | 6 | 300 |
| 08.02.2016 | 03.00 | 1.3 | 4 | 300 |
| 17.03.2016 | 17.30 | 1.6 | 4 | 650 |
| 17.03.2016 | 21.00 | 0.9 | 2 | 300 |
| 17.03.2016 | 22.00 | 0.9 | 4 | 170 |
| 03.10.2016 | 17.00 | 0.6 | 4 | 150 |
| 03.10.2016 | 19.00 | 1.6 | 4 | 600 |
| 27.10.2016 | 18.00 | 1.2 | 4 | 200 |
| 27.10.2016 | 21.00 | 1.35 | 5 | 500 |
| 20.01.2017 | 19.00 | 1.1 | 5 | 370 |
| 26.01.2017 | 19.30 | 1.1 | 4 | 400 |
| 27.01.2017 | 21.00 | 1.2 | 4.5 | 200 |
| 17.12.2017 | 20.30 | 1 | 4 | 400 |
| 18.12.2017 | 19.00 | 1.15 | 4 | 300 |
| **Substorms during PCPs** | | | | |
| 28.11.2011 | 22.30 | 1.25 | 6 | 200 |
| 18.01.2012 | 23.00 | 1.15 | 4 | 75 |
| 04.12.2013 | 19.00 | 0.9 | 4 | 150 |
| 15.12.2014 | 19.00 | 1.3 | 6 | 110 |
| 11.01.2016 | 19.30 | 1.5 | 2 | 400 |
| 11.01.2016 | 23.00 | 0.5 | 6 | 200 |


**Table 3. The values of the phase scintillation index, ROT, determined form GPS receiver at NYA station, and corresponding levels of the geomagnetic field disturbance (ΔX) during SSC/SI events.**


| Date (day/month/year) | Time, UT | Phase index, deg | ROT, TECU/min | ΔX, nT |
|---|---|---|---|---|
| **SSC/SI events** | | | | |
| 22.01.2012 | 06.20 | 0.4 | 4 | 300 |
| 24.01.2012 | 15.00 | 0.3 | 4 | 400 |
| 08.03.2012 | 11.00 | 0.35 | 3 | 200 |
| 14.07.2012 | 18.00 | 0.2 | 2.5 | 50 |
| 17.03.2013 | 06.00 | 0.8 | 3 | 550 |
| 02.10.2013 | 02.00 | 0.6 | 3 | 250 |
| 13.12.2013 | 13.30 | 0.3 | 2 | 50 |
| 12.09.2014 | 16.00 | 0.55 | 4 | 200 |
| 21.12.2014 | 19.30 | 0.7 | 7 | 80 |
| 07.01.2015 | 06.30 | 0.25 | 2 | 50 |
| 15.08.2015 | 08.30 | 0.3 | 2 | 400 |
| 14.12.2015 | 13.30 | 0.35 | 3 | 100 |
| 04.04.2017 | 04.40 | 0.5 | 3 | 70 |
| 07.09.2017 | 23.00 | 0.9 | 7 | 300 |




**Table 4. The values of the phase scintillation index, ROT, determined form GPS receiver at NYA station during PCPs events.**


| Date (day/month/ year) | Time, UT | Phase index, deg | ROT, TECU/ min |
|---|---|---|---|
| Polar cap patches | | | |
| 28.11.2011 | 17.00-22.00 | 0.4 | 8 |
| 29.11.2011 | 00.00-08.00 | 0.8 | 11 |
| 29.11.2011 | 14.00-22.00 | 0.4 | 6 |
| 30.11.2011 | 16.00-22.00 | 0.7 | 9 |
| 30.11.2011 | 10.00-14.00 | 0.5 | 7 |
| 18.01.2012 | 16.00-22.00 | 0.6 | 8 |
| 22.01.2012 | 08.00-12.00 | 0.9 | 6 |
| 22.01.2012 | 17.00-23.00 | 0.5 | 9 |
| 13.01.2013 | 17.00-22.00 | 0.6 | 10 |
| 04.11.2013 | 16.00-23.00 | 0.85 | 9 |
| 05.11.2013 | 20.00-24.00 | 0.35 | 6 |
| 06.11.2013 | 15.00-24.00 | 0.75 | 8 |
| 07.11.2013 | 06.00-12.00 | 0.8 | 10 |
| 04.12.2013 | 18.00-22.00 | 0.8 | 9 |
| 20.01.2014 | 08.00-10.00 | 0.9 | 3 |
| 21.01.2014 | 12.00-21.00 | 0.3 | 7 |
| 22.01.2014 | 06.30-15.00 | 0.8 | 7 |
| 22.01.2014 | 19.30-22.00 | 0.2 | 4 |
| 23.01.2014 | 19.00-22.00 | 0.4 | 7 |
| 24.01.2014 | 17.00-21.00 | 0.3 | 4 |
| 26.01.2014 | 18.00-24.00 | 0.3 | 2 |
| 28.01.2014 | 16.30-24.00 | 0.5 | 10 |
| 30.01.2014 | 19.00-22.00 | 0.3 | 5 |
| 18.12.2014 | 21.00-24.00 | 0.7 | 6 |
| 15.12.2014 | 09.00-24.00 | 0.4 | 7 |
| 16.12.2014 | 18.00-24.00 | 0.2 | 7 |
| 10.02.2015 | 20.00-24.00 | 0.4 | 15 |
| 11.02.2015 | 12.00-22.00 | 0.4 | 6 |
| 12.02.2015 | 15.00-23.00 | 0.3 | 7 |
| 13.02.2015 | 19.00-24.00 | 0.3 | 9.5 |
| 28.02.2015 | 17.00-23.00 | 0.35 | 11 |
| 20.03.2015 | 07.00-13.00 | 0.7 | 11 |
| 05.01.2016 | 12.00-18.00 | 0.4 | 5 |
| 07.02.2016 | 16.00-22.00 | 0.2 | 4 |
| 09.02.2016 | 19.00-23.00 | 0.2 | 4 |
| 09.02.2016 | 09.00-12.00 | 0.6 | 4 |
| 10.02.2016 | 18.00-23.00 | 0.2 | 4 |
| 12.02.2016 | 08.00-13.00 | 0.65 | 7 |