# Peer review of "Influence of different types of ionospheric disturbances on GPS signals at polar latitudes"

_Annales Geophysicae, 2020_

## Author Comment (AC2)

[revised manuscript text omitted]

---

## Author Response (AR1)

[revised manuscript text omitted]

During this event the strong ... scintillations (0.8 radians) w... whole time interval of the PC... Amplitude S4 index have no ... response. ¶
It was observed VTEC incre... 25 TECU) caused by the PC... ROT variations have high va... TECU/min). ¶
[Jin et al., 2017] investigated... scintillations around cusp re... that cusp precipitation has st... influence on GPS phase scin... it combined with the PCP. O... also confirm this finding. Th... January 2016 (Figure 1) was... while event on 7 November ... was accompanied by PCP. ◆

  <#> **Precipitations assoc...
  interplanetary shock arr...**
¶
It is also considered the exa... interplanetary shock influenc... scintillations of GPS signals ... example for the event of 22 ... presented. It is well known t... interplanetary shock interact... Earth's magnetosphere leads... particle precipitation into the... and leads to the appearance ... shock-aurora [Zhou et al., 20... paper [Belakhovsky et al., 20... that shock wave arrival leads... TEC increase at high latitude... The considered interplanetar... accompanied by an abrupt in... solar wind velocity (from 32... density (from 20 to 40 cm$^{-3}$) ... module of the interplanetary ... (from 12 to 25 nT) according... database, abrupt growth of th...

[revised manuscript text omitted]

plasma density according t
42m radar data; phase sci
and amplitude scintillation
according to the GPS recei
station; TEC and ROT var
according to the GPS recei
station; geomagnetic field
component) at NYA statio
January 2012.

We thank to the both referees for interest to our paper, careful reading and helpful comments and remarks. Below are our answers to the remarks.

**ANSWER TO REFEREE 1:**

**Referee 1:**
        «It is mentioned in the paper that 150 events have been considered, but there is only data from 5 events used in the paper. This raises doubt about whether the conclusion in the paper is generally valid, or only valid for this set of events. If 150 events have been analyzed, I would expect there to be at least some summary of the results included in the paper. It should include at least some key statistics derived from the analysis of the 150 events…»
**Answer:**
        To describe statistically our research we have added 4 tables and section 3.5 to the paper. These tables include 120 events: 33 cusp/daytime precipitations, 35 substorms, 14 SSC events and 38 PCPs. It is less than 150 events which were mentioned in previous variant of the paper. But the tables already are quite large. We have selected mainly interesting events. Of course we have considered much more event for the 2010-2017 years. But the paper cannot include all of these events. The data from the tables confirms our case studies.
        We have decided to remove Figure 8 because now it is too much Figures and Tables. At the same time the dayside PCPs were also observed during another considered event 22 January 2012. The characteristics of the 7 November 2013 event are presented now in the Table 4. We have changed numeration of the Figures.
        We also have combined Figures 5 and Figure 6 on same plot to better compare the level of GPS scintillations on Svalbard and in Scandinavia. Now it is Figure 7.

**Referee 1:**
«There are numerous language issues throughout the paper. While the errors are not critical for the understanding of the paper, it would be significantly improved by careful proof-reading».
**Answer:**
We have tried to remove the language issues.

**Referee 1:**
«Lines 84-91: You define the phase scintillation index twice. Fortunately, the definitions are the same. But you may want to delete one of the sentences».
**Answer:** We have corrected. Now we present only one definition of the phase scintillation index in the paper (Lines 105-107)

**Referee 1:**
«Line 125: The reference [Belakhovsky et al., 2019] does not match the reference list…»
**Answer:** We have corrected. Line 147.

**Referee 1:**
«Line 147: "Possibly it's due to the field of view of EISCAT radar not coincides with the field of view of GPS receivers." It is clear that the radar and the GNSS observations cannot fully overlap. But, is it possible to provide any more assessment of the degree of overlap?...
…for the case studies, have you checked which of the satellites (if any) actually passed through the region of the ionosphere that is observed by EISCAT?».
**Answer:** To show the degree of overlap of GPS satellites and EISCAT 42m radar we have plotted on Figure 4 the ionosphere projections of the GPS satellites, location of EISCAT radar and location of GPS receiver at NYA. Of course for the large scale disturbances (300-500 km and more) which was considered in the paper this overlap is not significant because several

ionosphere projection points of GPS satellites (2-4) are always located near field of view of the EISCAT radar. So practically we have continuous measurements of the ionosphere parameters determined by the NYA GPS receiver.
Lines 190-192.

**Referee 1:**
«Lines 187-188:
"The PCP is also identified in the aurora intensity variations as forms propagating from the polar to low latitudes in 630.0 nm (red line) emission (Figure 7) at 19.00-23.00 UT according to LYR all-sky camera observations."
The keogram seems to show multiple patches, not just one.
Are you using the abbreviation "PCP" to refer to a multitude of patches, instead of just one patch?... ».
**Answer:** It is corrected. Now we use abbreviation "PCPs" everywhere in the paper.

**Referee 1:**
«Lines 191-192: "At latitudes of SKN (TRO) stations the PCP manifests itself as a long lasting substorm (more than 4 hours duration) with the amplitude 200-250 nT."
A patch is not the same as a substorm…»
**Answer:** Yes, we agree. We rewrote this sentence.
Lines 234-235:
«During appearance of PCPs near the Svalbard at latitudes of the SKN (TRO) stations a long lasting substorm (more than 4 hours duration) with the amplitude 200-250 nT is observed».
Line 339-340:
«Comparison of the EISCAT observations on Svalbard and in Tromso shows that during PCPs appearance on Svalbard a typical substorm at lower latitudes (Tromso) was observed. The level of the phase scintillations are quite comparable at high (Tromso) and polar (Svalbard) latitudes».

**Referee 1:**
«Line 195 (Daytime polar cap patches): For a CIR event, I am interested to know the approximate solar wind properties. In particular, what was the maximum value of the solar wind speed. It is not required to include a plot of the solar wind data, just state the value in the text».
**Answer:** The solar wind speed for this event was about 390 km/s according to the OMNI database. But we have decided to remove this event as mentioned above due to large numbers of Figures and Tables. The characteristics of the 7 November 2013 event are presented now in the Table 4.

**Referee 1:**
«Lines 204-205: "cusp precipitation has stronger influence on GPS phase scintillation when it combined with the PCP. Our analyses also confirm this finding." You stated that "PCPes were registered in time interval 06.00-12.00 UT".
…
You have not stated the presence of precipitation at any time interval for this event. To reach your conclusion, did you make assumptions regarding the particle precipitation also for times when EISCAT did not directly observe precipitation? If so, please state the assumptions clearly in the text».
**Answer:** Yes, you are right. For the event 7 November 2013 there is no particle precipitation during PCPs appearance. We remove Figure 8 due to large number of Figures and Tables.
We rewrite this sentence (Line 278):
« [Jin et al., 2017] investigated the GPS scintillations around cusp region and found that cusp precipitation has stronger influence on GPS phase scintillation when it combined with the PCPs.

Our analyses show that daytime PCPs can produce stronger GPS phase scintillations than cusp/dayside precipitations».

**Referee 1:**
«Line 217: "module of the interplanetary magnetic field". Do you mean "magnitude of the interplanetary magnetic field"?».
**Answer:** Yes, we mean magnitude of the interplanetary magnetic field. It is corrected. Line 213.

**Referee 1:**
«Line 219: Please spell out the abbreviation "SSC" in full at its first occurrence.
**Answer:** We have indicated (Line 209).

**Referee 1:**
«Lines 243-244: "Possibly low values of amplitude scintillations at high latitudes are caused by the low elevation angles of GPS satellites at these regions". Please explain how observing at low elevation decreases the amount/magnitude of amplitude scintillation».
**Answer:** This is only hypothesis. The plasma irregularities producing high-latitudes scintillations mainly formed along the geomagnetic field. At polar latitudes (near Svalbard) the geomagnetic field is close to vertical. So radiowave beam of GPS satellite penetrate through the ionosphere not along geomagnetic field. If we will have satellite with higher inclination angle the amplitude scintillation possibly can be detected. But this hypothesis needs to be tested. Line 309.

**ANSWER TO REFEREE 2:**

**Referee 2:**
«My main concern is about the meaning of the phase scintillation index…»
**Answer:**
      We agree with the referee that the problem of the GNSS signal detrending for the high-latitude disturbances is actual, the fixed cut-off frequency is not always able to remove the refractive (deterministic) effects at high latitudes. At the same time the choice of the optimal cut-off frequency is still open problem.

      There are a lot of papers and results since [Fremouw et al., 1978] where the standard cut-off frequency (0.1 Hz) was used. So the classic definition of the phase and amplitude scintillation indexes already is established in literature.

      The best way in this situation in our opinion is to introduce new scintillations indexes. Therefore, there will be clarity through the researchers. For example [Mushini, 2012] introduced improved phase-scintillation index, $\sigma_{chain}$, [Forte, 2005] introduced S$\varphi$ index.

      We kept our events and we used the standard cut-off frequency (0.1 Hz) for the detrending. We mentioned in the paper about problem of the GPS signal detrending, we wrote that the term "phase scintillation index" used in our study means the phase fluctuations due to the presence of large-scale irregularities.

This text we pasted to the Introduction (Lines 41-52):
«The phase scintillations index is determined mainly by the diffractive (stochastic) GPS signal variations. In order to calculate scintillation indices a long-term trend caused by the satellite motion in relation to the receiver and ionosphere changes needs to be removed. A standard cutoff frequency (0.1 Hz) is commonly used for signal detrending since (Fremouw et al., 1978). This cutoff frequency is adequate for the equatorial and midlatitude ionosphere. But the high-latitude ionosphere is characterized by the high and variable ionospheric drift velocity (~100 m/s–1 500 m/s) which add refractive variations to the phase scintillation index. The value of the cutoff frequency affects on the phase scintillation index. So it often leads to the strong phase scintillations without amplitude scintillation. Some researchers to solve this problem introduce new scintillations indexes: Mushini et al. (2012) introduced improved phase-scintillation index ($\sigma_{chain}$), Forte (2005) introduced S$\varphi$ index. The fast iterative filtering signal decomposition technique was used to find optimal cutoff frequency (Ghobadi et al., 2020; Spogli et al., 2021). De Franceschi et al. (2009) have demonstrated the difficulty in finding the optimum cutoff frequency for statistical studies and they proposed to look for actual scintillations investigating the simultaneous occurrence of S4 and $\sigma_\varphi$ increase».

This text we pasted to the chapter "2. Instruments" (Lines 110-113):
« In our paper we use the standard detrending (0.1 Hz) in order to be able to compare our results with previous results obtained with using this GPS receiver on Svalbard (Clausen et al., 2016; Jin et al., 2015, 2016, 2017, 2018). The choice of the fixed cutoff frequency can add refractive variations to the phase scintillation index. So term "phase scintillation index" used in our study means the phase fluctuations due to the presence of large-scale irregularities (above the Fresnel radius)».

We add some papers to the References:
- De Franceschi G., Spogli L., Alfonsi L., Romano V., Cesaroni C., Hunstad I. The ionospheric irregularities climatology over Svalbard from solar cycle 23, Scientific reports, 9, 1, 1-14, https://doi.org/10.1038/s41598-019-44829-5, 2019.

- Forte B. Optimum detrending of raw GPS data for scintillation measurements at auroral latitudes, Journal of Atmospheric and Solar-Terrestrial Physics, 67, 1100–1109, https://doi.org/10.1016/j.jastp.2005.01.011, 2005.
- Fremouw E.J., Leadabrand R.L., Livingston R.C., Cousins M.D., Rino C.L., Fair B.C., Long R.A. Early results from the DNA wideband satellite experiment — complex-signal scintillation, Radio Science, 13(1), 167–187, https://doi.org/10.1029/RS013i001p0016, 1978.
- Ghobadi H., Spogli L., Alfonsi L., Cesaroni C., Cicone A., Linty N., Romano V., Cafaro M. Disentangling ionospheric refraction and diffraction effects in GNSS raw phase through fast iterative filtering technique, GPS Solutions, 24, 1-13, https://doi.org/10.1007/s10291-020-01001-1, 2020.
- Mushini S.C., Jayachandran P.T., Langley R.B., MacDougall J.W., Pokhotelov D. Improved amplitudeand phase-scintillation indices derived from wavelet detrended high-latitude GPS data, GPS Solutions, 16, 3, 363-373, https://doi.org/10.1007/s10291-011-0238-4, 2012.
- Spogli L., Ghobadi H., Cicone A., Alfonsi L., Cesaroni C., Linty N., Romano V., and Cafaro M. Adaptive Phase Detrending for GNSS Scintillation Detection: A Case Study Over Antarctica, IEEE Geoscience and Remote Sensing Letters, https://doi.org/10.1109/LGRS.2021.3067727, 2021.

**Referee 2:**
«Are you applying an elevation mask to minimize the multipath?
Did you projected the scintillation indices to the vertical to minimize the geometrical effects as suggested by Spogli et al. (2013)»
**Answer:**
We did not apply an elevation mask to minimize the multipath. We did not project the scintillation indices to the vertical to minimize the geometrical effects.
We use the data from the same GPS receiver on Svalbard that was used in papers published before. So we apply the same data processing in order to be able to compare our results with results in other papers published in rating journals.

- Clausen L. B. N., Moen J. I., Hosokawa K., Holmes J. M.: GPS scintillations in the high latitudes during periods of dayside and nightside reconnection, J. Geophys. Res., 121, 3293–3309, https://doi.org/10.1002/2015JA022199, 2016.
- Jin Y., Moen J. I., and Miloch W. J.: On the collocation of the cusp aurora and the GPS phase scintillation: A statistical study, J. Geophys. Res., 120, 9176–9191, https://doi.org/10.1002/2015JA021449, 2015.
- Jin Y., Moen J. I., Miloch W. J., Clausen L. B. N., and Oksavik K.: Statistical study of the GNSS phase scintillation associated with two types of auroral blobs, J. Geophys. Res., 121, 4679–4697, https://doi.org/10.1002/2016JA022613, 2016.
- Jin Yaqi, Zhou Xiaoyan, Moen Jøran I., Hairston Marc. The auroral ionosphere TEC response to an interplanetary shock // Geophysical Research Letters, Vol. 43, Issue 5, pp. 1810-1818. https://doi.org/10.1002/2016GL067766. 2016.
- Jin Y., Moen J. I., Oksavik K., Spicher A., Clausen L. B.N., Miloch W. J. GPS scintillations associated with cusp dynamics and polar cap patches: J. Space Weather Space Clim., 7, A23. https://doi.org/10.1051/swsc/2017022, 2017.
- Jin Y., Oksavik K. GPS scintillations and losses of signal lock at high latitudes during the 2015 St. Patrick's Day storm, J. Geophys. Res., 123, 7943–7957, https://doi.org/10.1029/2018JA025933, 2018.
- Jin Y., Moen J.I., Spicher A., Oksavik K., Miloch W. J., Clausen L. B. N., et al. Simultaneous rocket andscintillation observations of plasmairregularities associated with a reversed flow event in the cusp ionosphere, Journal of Geophysical Research, 124, 7098–7111. https://doi.org/10.1029/2019JA026942. 2019.

- Chernyshov A.A., Miloch W.J., Jin Y., Zakharov V.I. Relationship between TEC jumps and auroral substorm in the high-latitude ionosphere, Scientific Reports. 10:6363. https://doi.org/10.1038/s41598-020-63422-9. 2020.

**Referee 2:**
Line 45: "…in the magnetosphere tail and then **released** into the auroral ionosphere"
**Answer:**
It is corrected (Line 61)

**Referee 2:**
Lines 51-52: "At polar latitudes polar cap patches **can** produce severe ionosphere disturbances."
**Answer:**
It is corrected (Line 69).

**Referee 2:**
Lines 195-196: Reword the sentence "It was also analyzed the influence of GPS signals scintillations the polar cap patches propagating on the dayside".
**Answer:**
It is corrected (Line 241-242):
«It was also analyzed the influence of polar cap patches which was observed on the dayside on GPS signals scintillations».

**Referee 2:**
Lines 243-244: "Possibly low values of amplitude scintillations at high latitudes are caused by the low elevation angles of GPS satellites at these regions."
The issue is the opposite! When the elevation is low the S4 could be higher because of the contributions from longer path from the transmitter to the receiver.
**Answer:**
This is hypothesis. The plasma irregularities producing high-latitudes scintillations mainly formed along the geomagnetic field. At polar latitudes (near Svalbard) the geomagnetic field is close to vertical. So radiowave beam of GPS satellite penetrate through the ionosphere not along geomagnetic field. If we will have satellite with higher inclination angle the amplitude scintillation possibly can be detected. But this hypothesis needs to be tested.

---

## Referee Report (RR1)

2nd Review of "Influence of different types of ionospheric disturbances on GPS signals at polar latitudes"

Most of the reviewers' comments have been adequately answered, but there are some remaining issues that need to be solved.

On lines 168-170 you write:
 *"The growth of the phase index was seen not during all period of Ne enhancement (100-200 km) measured by the EISCAT radar. Possibly this is due to the field of view of EISCAT radar not coincides with the field of view of GPS receivers. So for 170 small-scale and medium-scale Ne increase is not always correlate with phase index growth."*

And on lines 190 - 192 you write:
 *"We have plotted ionosphere pierce points (IPP) to show how it overlaps with the EISCAT 42m and all-sky camera field of view (Figure 4). For large-scale disturbances (hundreds of km) which were considered in this paper this overlap does not matter. Always some GPS satellites can be used to register the ionosphere disturbance."*

This seems like a slight contradiction to me. Maybe I have just misunderstood something?
In the second piece of text you say that there are always some GPS satellites that can register the disturbance, but in the first piece of text you say that there are no GPS satellites that register the disturbance. Please clarify.

Lines 307 - 309:
 *"Possibly low values of amplitude scintillations at high latitudes are caused by the low elevation angles of GPS satellites at these regions. Since irregularities producing amplitude scintillations can be formed in
the field-aligned direction. But this hypothesis needs to be tested."*

I understand what you are saying, but I think your hypothesis will quickly run into problems.
Plot the S4 index as a function of elevation, and you will typically see increased values at low elevations.
See for example the plot on the following page, which is a time series of S4 for an entire pass of a satellite observed by a scintillation receiver at Ny-Ålesund.
The start and end of the time series, which are at the lowest elevation, have the highest values.
You will need to explain how this can be consistent with your hypothesis.

[Figure]

According to scintillation theory, one important factor for the amplitude of S4 is the distance from the irregularity layer to the observer.

See for example Eq. 4 and 6 and associated paper text in "On the Relationship Between the Rate of Change of Total Electron Content Index (ROTI), Irregularity Strength (CkL), and the Scintillation Index (S4)" by Carrano et al.

(https://agupubs.onlinelibrary.wiley.com/doi/10.1029/2018JA026353)

This is an important part of the explanation of the differences between low latitude and high latitude scintillation, as the generating mechanism are different (including being typically located at different altitudes).

---

## Author Response (AR2)

We thank to the referee for the careful reading and helpful comments and remarks. Below are our answers to the remarks.

**Referee:**
«This seems like a slight contradiction to me. Maybe I have just misunderstood something?
In the second piece of text you say that there are always some GPS satellites that can register the disturbance, but in the first piece of text you say that there are no GPS satellites that register the disturbance. Please clarify».

**Answer:**
In the first piece we discuss the small-scale ionosphere disturbances, i.e. with the scale size lower than overlap between the EISCAT and NYA GPS receiver fields of view. At some moments these ionosphere disturbances can be registered by EISCAT and cannot be registered by the GPS satellites. So for some time intervals we see the growth of the ionosphere plasma density and we can't see the growth of the phase scintillation index.
In the second piece we discuss the large-scale ionosphere disturbances. Large-scale ionosphere disturbances can be registered by the several GPS satellites.

**Referee:**
«I understand what you are saying, but I think your hypothesis will quickly run into problems.
Plot the S4 index as a function of elevation, and you will typically see increased values at low elevations. See for example the plot on the following page, which is a time series of S4 for an entire pass of a satellite observed by a scintillation receiver at Ny-Ålesund. The start and end of the time series, which are at the lowest elevation, have the highest values. You will need to explain how this can be consistent with your hypothesis…».

**Answer:**
We agree with the referee that our hypothesis is not obvious and give raise a lot of questions. At the same time, this hypothesis does not affect on results obtained in the paper. So we decide to remove this hypothesis from the paper.